

# Packaging style design based on visual semantic segmentation technology and intelligent cyber physical system

Jiahao Wang

College of Art and Design, Xi'an Mingde Institute of Technology, Xi'an, China

Corresponding author
Jiahao Wang, 15109257536@163.com

## ABSTRACT

The integration of image segmentation technology into packaging style design significantly amplifies both the aesthetic allure and practical utility of product packaging design. However, the conventional image segmentation algorithm necessitates a substantial amount of time for image analysis, rendering it susceptible to the loss of vital image features and yielding unsatisfactory segmentation results. Therefore, this study introduces a novel segmentation network, G-Lite-DeepLabV3+, which is seamlessly incorporated into cyber-physical systems (CPS) to enhance the accuracy and efficiency of product packaging image segmentation. In this research, the feature extraction network of DeepLabV3 is replaced with Mobilenetv2, integrating group convolution and attention mechanisms to proficiently process intricate semantic features and improve the network's responsiveness to valuable characteristics. These adaptations are then deployed within CPS, allowing the G-Lite-DeepLabV3+ network to be seamlessly integrated into the image processing module within CPS. This integration facilitates remote and real-time segmentation of product packaging images in a virtual environment.Experimental findings demonstrate that the G-Lite-DeepLabV3+ network excels at segmenting diverse graphical elements within product packaging images. Compared to the original DeepLabV3+ network, the intersection over union (IoU) metric shows a remarkable increase of 3.1%, while the mean pixel accuracy (mPA) exhibits an impressive improvement of 6.2%. Additionally, the frames per second (FPS) metric experiences a significant boost of 22.1%. When deployed within CPS, the network successfully accomplishes product packaging image segmentation tasks with enhanced efficiency, while maintaining high levels of segmentation accuracy.

## INTRODUCTION

Packaging serves as a highly intuitive means to evoke the desire to purchase among consumers. By interacting with product packaging, consumers gather essential information about the product and make purchasing decisions (*Frierson, Hurley & Kimmel, 2022*). Consequently, it is crucial that the design style of product packaging aligns with consumers' consumption psychology and is easily recognizable (*He, 2022*). Modern product packaging heavily relies on graphic elements as a primary design approach (*He, 2022*). This design

style can forgo textual descriptions and instead rely on interesting and vibrant graphic elements to convey product information in a more intuitive manner, swiftly attracting buyers and boosting product sales (*Qiao & William, 2021*; *Xu, 2022*). Given the intensifying competition within the product market and the continuous advancements in computer image technology, the integration of image segmentation technology into product packaging design can optimize the visual cognitive impact of packaging, incite customers' purchase desire, and enhance sales. Thus, it becomes imperative to explore effective and accurate segmentation techniques for the graphic elements and design styles present in product packaging images.

Image segmentation involves dividing an image into distinct regions based on similar properties. With the advancement of deep learning technology, image segmentation has witnessed rapid progress and widespread application across various industries (*Kumar, Kumar & Lee, 2022*). For instance, *Kumar, Kumar & Lee (2022)* proposed an efficient conduction neural network for semantic feature segmentation. *Li, Chen & Zhang (2019)* introduced a U-Net network structure, which is more suitable for fine image processing compared to the fully convolutional network (FCN). U-Net employs a combination of up-sampling and down-sampling to gradually obtain high-level semantic information. It also utilizes skip connections to concatenate feature maps of the same channel, enabling feature fusion and significantly enhancing segmentation performance. *Duan et al. (2018)* designed a lightweight Seg-Net model, incorporating a novel up-sampling method that optimizes memory usage and achieves more efficient image segmentation. Building upon U-Net, the U-Net++ network makes further advancements in image segmentation technology. It enhances the extraction of feature information at different levels through pruning, while addressing the issue of self-adaptive sampling depth among different samples. However, it suffers from a sudden increase in model parameters, resulting in higher computational costs (*Zhou et al., 0000*). In line with the progression of network models, *Tan et al. (2021)* proposed an ACU-Net-based image segmentation method, which utilizes depth separable convolution to reduce model parameters. *Re, Stanczyk & Mehrkanoon (2021)* introduced an attention mechanism to U-Net, replacing traditional convolution with depth-wise convolution. *Cao & Zhang (2020)* presented an improved ResU-Net model for high-resolution image segmentation, effectively achieving segmentation at a higher resolution. *He, Fang & Plaza (2020)* proposed a mixed attention network for accurate building segmentation. *Zhao et al. (2022)* devised an image segmentation algorithm based on Inceptionv3, while Shi et al. (*Zhao et al., 2017*) developed a pyramid scene analysis network that integrates context information to perform semantic segmentation of scene objects. *He et al. (2017)* introduced Mask R-CNN, which achieves high-quality semantic segmentation while also performing object detection. DeepLabv1 (*Wu et al., 2022*) enhances boundary details through random post-processing of FCN segmentation results. DeepLabv2 (*Ji et al., 2020*) replaces up-sampling with dilated convolution and introduces a hole pyramid module to reduce computational complexity and error rates. DeepLabv3 (*Rogelio et al., 2022*) further optimizes the spatial pyramid module to capture multi-scale information more effectively. DeepLabv3+ (*Zhou, 2022*) incorporates an encoder–decoder structure based on DeepLabv3, facilitating superior feature fusion. PSPNet (*Gan, 2022*) introduces a

pyramid pooling module to fully leverage context information. However, while these image segmentation methods yield improved results in various contexts, they exhibit limitations such as poor adaptability, high memory consumption, and inadequate segmentation of detailed features, rendering them unsuitable for segmenting product packaging images. The requisites for image segmentation algorithms extend beyond solely enhancing segmentation accuracy; they encompass rational resource allocation and optimization of algorithm efficiency. In light of these requirements, cyber-physical systems (CPS), as intelligent systems integrating algorithms, networks, and physical entities, have garnered the attention of researchers in the field (*Maru et al., 2022*). The algorithm embedded within CPS facilitates remote, reliable, safe, cooperative, and intelligent algorithm control through diverse real-time modules. CPS exhibits substantial potential for wide-ranging applications in electric power, petrochemicals, and medical treatment (*Xu et al., 2022*). Consequently, deploying the image segmentation algorithm within CPS offers enhanced support for algorithm regulation and application, minimizing training and testing duration, thereby enabling the completion of more intelligent and efficient product packaging image segmentation tasks.

To address the challenges posed by inaccurate segmentation of graphic elements in product packaging images and low segmentation efficiency encountered by traditional image segmentation algorithms, this article proposes a G-Lite-DeepLabV3+ segmentation network and deploys it within CPS to accomplish precise and efficient product packaging image segmentation. The key innovations are as follows: (1) Replacing the DeepLabV3+ network's backbone with MobileNetV2, resulting in a streamlined network structure, accelerated feature extraction, and overfitting prevention. (2) Introducing group convolution to supplant traditional convolution in MobileNetV2 and the space pyramid module while eliminating the batch specification layer to reduce network complexity. (3) Integrating an attention module following the space pyramid module to enhance the network's recognition rate. (4) Deploying the G-Lite-DeepLabV3+ segmentation network within CPS to further enhance segmentation efficiency.

The article's structure is as follows: The second section exposes the G-Lite-DeepalV3+-based product packaging image segmentation network. In the third section, the deployment of CPS is elucidated. Section 4 showcases the segmentation performance of this method through experimental demonstrations. The fifth section encompasses a comprehensive summary and a forward-looking perspective on the content presented in this article.

# IMAGE SEGMENTATION NETWORK OF PRODUCT PACKAGING BASED ON G-LITE-DEEPLABV3+

This article have chosen to augment the Deeplabv3+ network, renowned for its exemplary semantic segmentation capabilities, streamlined architecture, and expedited segmentation speed. Considering the original backbone network's proneness to overfitting and lackluster performance in the assigned task, the authors propose replacing it with the MobileNetv2 network. The MobileNetv2 network has undergone subtle refinements to alleviate overfitting concerns and bolster feature utilization. Moreover, an attention module

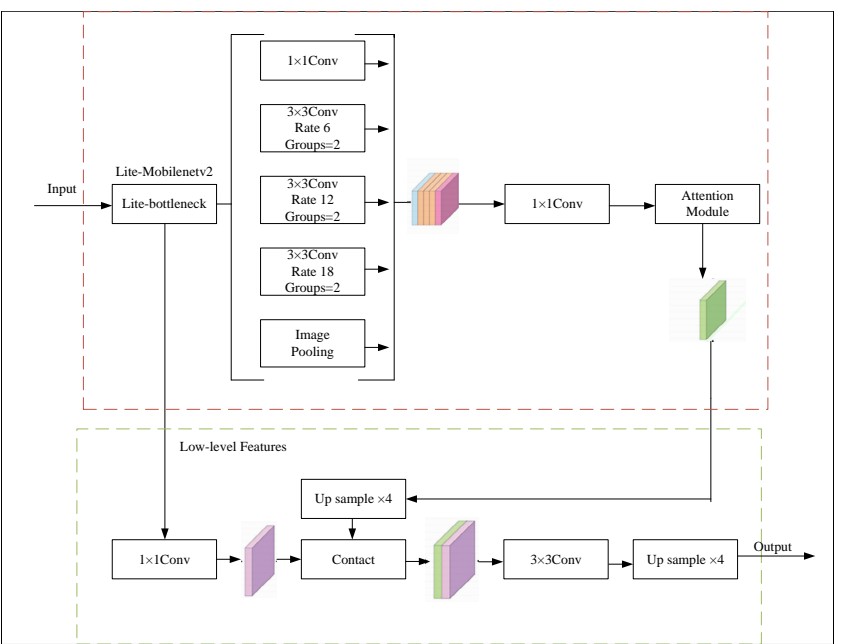

**Figure 1** G-lite-Deeplabv3+ structure.

is introduced subsequent to the hollow pyramid structure to heighten the decoder's receptiveness to the target region. Consequently, the decoder amalgamates the feature map and conducts pixel classification. The schematic representation of the network architecture can be observed in Fig. 1.

## Group convolution

This article replaces the ordinary convolution in MobileNetv2 and spatial pyramid structure with the block convolution. The grouping operation is to group the feature maps first and then perform the convolution operation when convolving the feature maps, and its principle is shown in Fig. 2. If the characteristic graph size of an ungrouped network is $C \times W \times H$, the number of convolution kernel groups is $N$, convolution kernel size is $C \times K \times K$. To output N groups of characteristic maps, $C \times K \times K \times N$ parameters need to be learnt, if the characteristic group is divided into $G$ groups, only $(C/G) \times K \times K \times N$ parameters need to be learnt, The total number of parameters is reduced to $\frac{1}{G}$. In addition, the group convolution can also be regarded as a dropout of the original feature graph to avoid over-fitting.

## Lightweight backbone

The backbone of the new network adopts MobileNetv2 as the main network structure, and the specific structure is shown in Table 1, where $H^2$ represents the number of pixels of the input image; $C$ represents the number of channels; $t$ represents the multiplication factor; $n$ represents the number of repetitions; $s$ represents the convolution step size when each Lite-bottleneck module is repeated for the first time.

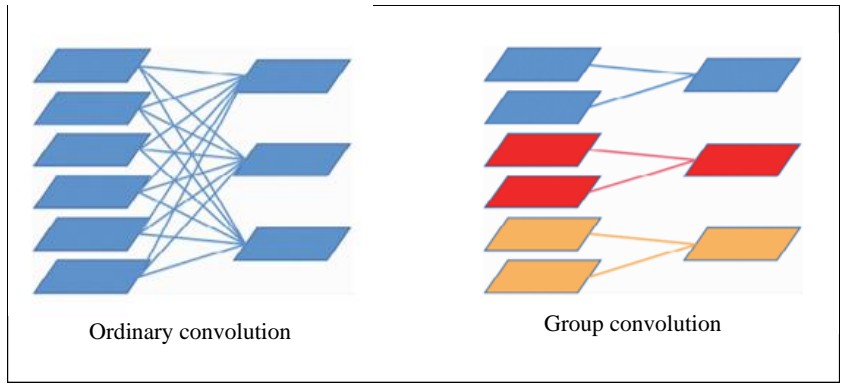

Ordinary convolution            Group convolution

**Figure 2** **Schematic diagram of group convolution.**

**Table 1** **G-lite-Deeplabv3+ network structure.**

| Input size | Operator | $t$ | $c$ | $n$ | $s$ |
|---|---|---|---|---|---|
| 512×512×3 | Conv2d | / | 32 | 1 | 2 |
| 256×256×32 | Lite-bottleneck | 6 | 16 | 1 | 1 |
| 256×256×16 | Lite-bottleneck | 6 | 24 | 2 | 2 |
| 128×128×24 | Lite-bottleneck | 6 | 32 | 3 | 2 |
| 64×64×32 | Lite-bottleneck | 6 | 64 | 4 | 2 |
| 32×32×64 | Lite-bottleneck | 6 | 96 | 3 | 1 |
| 32×32×96 | Lite-bottleneck | 6 | 160 | 3 | 2 |
| 16×16×160 | Lite-bottleneck | 6 | 320 | 1 | 1 |

Within this framework, the Lite-bottleneck represents a lightweight refinement of the original bottleneck proposed in this manuscript. It adheres to the same fundamental principle as the original bottleneck and can be subdivided into dimension-increasing, convolution, and dimension-decreasing layers. Notably, the activation functions employed in the dimension-increasing and convolution layers are ReLU. However, to preserve the integrity of the compressed features and prevent the ReLU function from compromising them, the activation functions within the "dimension-reducing layer" are linear functions, as visually depicted in Fig. 3. The key distinction lies in Lite-bottleneck's substitution of conventional convolution in the "dimension-increasing layer" and the "dimension-decreasing layer" with group convolution utilizing two groups, thereby reducing the parameter count.

Concurrently, in order to mitigate concerns regarding overfitting in external networks, this article have omitted the batch specification layer within the Lite-bottleneck, thereby reducing computational overhead. Within the phase of feature extraction, specific feature maps are derived as low-level semantic features, subsequently utilized as input for the decoder. The ultimate feature maps are directed towards the spatial pyramid pool structure. The intricate algorithm of the network is delineated in Table 2.

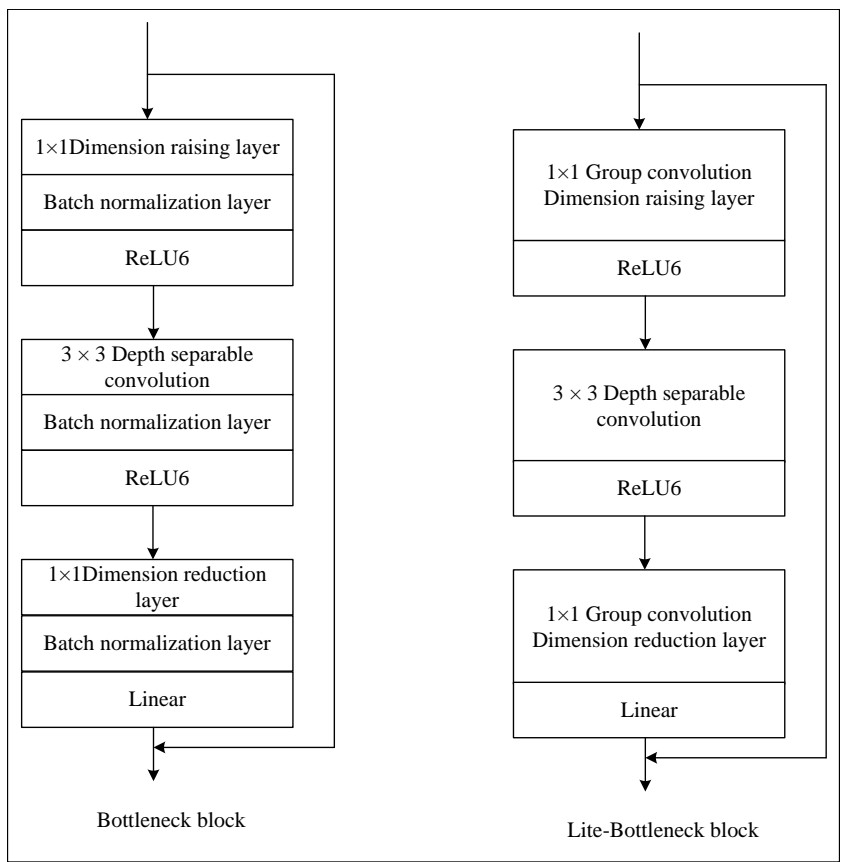

**Figure 3** Comparison of original bottleneck and Lite-bottleneck structures.

In comparison to the original Xception backbone network, the proposed structure significantly diminishes network depth and parameters. Consequently, it is better suited for the real-time segmentation task of product packaging images addressed in this article.

## Lightweight spatial pyramid pooling module

Within the DeepLabv3+ framework, the spatial pyramid structure assumes a pivotal role in extracting contextual information across multiple scales, thereby bolstering target detection and achieving more precise segmentation. By subjecting the image to pooling *via* the spatial pyramid, a comprehensive feature map is obtained, containing rich information. In order to expedite the segmentation process and enable real-time functionality, this study introduces a lightweight hollow space pyramid pool module.

To accomplish this, the cavity convolution is substituted with a cavity grouping convolution comprising two groups. This replacement effectively reduces the parameter count and fulfills the function of the batch specification layer. Furthermore, the feature maps produced by the lightweight hollow space pyramid pool modules are fused together and subsequently fed into an attention module.

**Table 2  Lightweight MobileNetv2 network algorithm.**

| Input image $X$ |
|---|
| $Xc = Conv(X)$ |
| $XR = ReLU(Xc)$ |
| for $n = 1$ to $N$ : |
| $X1 = ExpandedConv(Xn, groups = 2)$ |
| $X2 = DepthwiseConv(X1)$ |
| $X3 = ProjectConv(X2, groups = 2)$ |
| if $n = 2$: |
| $XLowLevel - Feature = Xn$ |
| end |
| $OutputXLowLevel - Feature, Xn$ |

The attention module fine-tunes the network's focus by assigning distinct weights to pixels, thereby augmenting both segmentation effectiveness and efficiency. In this article, the attention module consists of a sequence of a channel attention module and a spatial attention module. The channel attention module facilitates the extraction of more profound features, while the spatial attention module captures global information within images of product packaging.

Figure 4 shows the structure of channel attention. First, a feature graph of size $h' \times w' \times c_1$ is input, and it is changed to $h \times w \times c_2$ by the convolution transformation in the graph. Then, the feature map is compressed into $1 \times 1 \times c$ vector by global averaging pooling. The weight of each channel is obtained to get a channel attention module.

Figure 5 shows the structure of spatial attention. After the feature is input, the target in the image is cut, rotated, scaled and translated by the positioning network. To realize the above transformation of the target, the coordinate values of the target $(x, y)$ is taken as a two-row and one-column matrix, and it is multiplied by a two-row and two-column matrix, and then add it to a matrix of two rows and one column (the parameters in the matrix from top to bottom are $e$ and $f$). After the positioning network completes the above transformation of the target, the coordinates of the target image are transformed to the coordinates of the original image through grid generation. After that, the sampler is used to solve the gradient descent problem when the coordinates have decimals. Location network, grid generator and sampler constitute a complete spatial attention module.

## Loss function

The commonly used cross entropy function cannot deal with the phenomenon of product packaging image category imbalance and can't effectively monitor the network, leading to low segmentation performance. Therefore, the Dice coefficient difference function is used to train the network, which can punish the prediction with low confidence. A smaller Dice coefficient difference function will be obtained if the confidence is high. Formula (1) shows its calculation.

$$s = \frac{2|X \cap Y|}{|X| + |Y|} \tag{1}$$

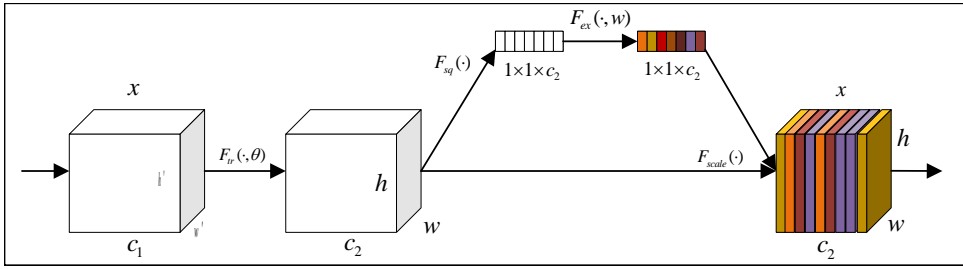

**Figure 4** Channel attention module.

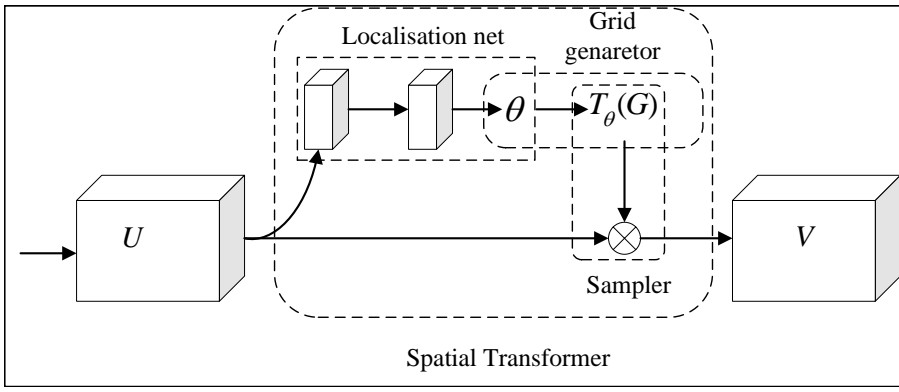

**Figure 5** Spatial attention module.

where $X$ is a label image; $Y$ is the forecast output graph; $|X \cap Y|$ is the sum of the point multiplication between the prediction graph and the partition graph and the addition of the element results of the results. The quantitative calculation of $X$ and $Y$ can use simple element addition, and finally obtain the Dice coefficient difference function, as shown in Formula (2).

$$L_{Dice} = 1 - \frac{2|X \cap Y|}{|X| + |Y|}. \tag{2}$$

## CPS DEPLOYMENT

CPS, as a complex system encompassing computing, networking, and physical entities, enables the interaction between algorithms and physical processes through a human-machine interface. This enables remote, reliable, real-time, safe, cooperative, and intelligent algorithm manipulation. In order to achieve more efficient product packaging image segmentation, this article integrates the G-Lite-DeepalBV3+ segmentation network into CPS, as depicted in Fig. 6.

Image acquisition module: this module captures the product packaging image and converts it into a computer-readable data stream. It includes components such as a camera, photo light source, and image acquisition card.

PeerJ Computer Science ______________________________

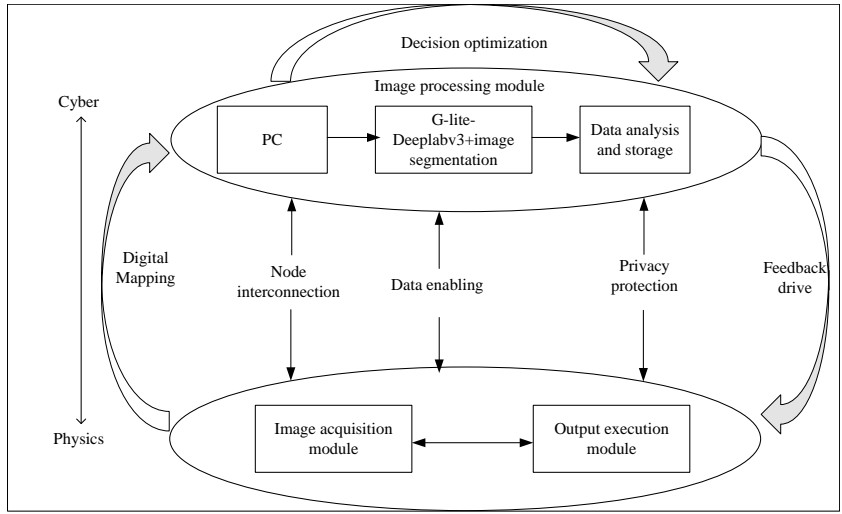

**Figure 6   CPS structure.**

Image processing module: this module comprises a PC, G-Lite-DeepalV3+ image segmentation algorithm, and storage and analysis capabilities. Initially, the PC receives the segmentation image and instructions. Then, the G-Lite-DeepalV3+ algorithm performs image segmentation on the received image. Finally, the segmentation result is stored, analyzed, and fed back to the output execution module.

Output execution module: The PLC (programmable logic controller) is chosen as the executive terminal controller to facilitate remote algorithm control, image segmentation, and result output under human supervision.

## EXPERIMENT AND ANALYSIS

The graphics card used in this experiment was an NVIDIA RTX 2070, the CPU was an AMD Ryzen 5 2600x, with 32 GB of memory, and the deep learning framework was Pytorch 1.3. During the training process, the picture was in JPG format and the label was in PNG format. The input initial learning rate was set to 0.00001, the weight attenuation value was 0.000001, and the super parameter was 0.9. RMSprop algorithm (*Jian, 2022*) was used to optimize each parameter, and the formula follows.

$$Sdw = \beta Sdw + (1 - \beta)dw^2 \tag{3}$$

$$Sdb = \beta Sdb + (1 - \beta)db^2 \tag{4}$$

$$w = w - \alpha \frac{dw}{\sqrt{Sdw + \varepsilon}} \tag{5}$$

$$b = b - \alpha \frac{db}{\sqrt{Sdb + \varepsilon}}. \tag{6}$$

**Table 3  Comparison of MIoU parameters.**

| Network name | Graphic element | background | amount to |
|---|---|---|---|
| Deeplabv3+ | 0.818 | 0.995 | 0.906 |
| Ours | 0.882 | 0.997 | 0.939 |

Among them, *Sdw* and *Sdb* represents the weight *w* and the gradient momentum of offset value *b* in iteration, $\alpha$ represents the learning rate, $\beta$ represents a superparameter, $\varepsilon$ is used to prevent the denominator from being zero.

The product packaging image data set constructed in this article was obtained from Baidu, Weibo and other websites, with a total of 500 color pictures. Each image contained different styles of text and graphic elements. The high-resolution image was cut into several images with the size of $256 \times 256$ pixels, and the images were manually selected with rich style information to enhance the data. After data enhancement, a total of 2,500 product packaging pictures were obtained.

## Experimental results of segmented images

To assess the segmentation recognition effectiveness of the algorithm, the experiment utilizes the product packaging images from the dataset as inputs and evaluates the corresponding image segmentation results. In this experiment, the training model predicts 100 images, and then the model's performance is evaluated by comparing the dataset and calculating the mean intersection over union (MIoU) parameter (*Ji et al., 2020*). The MIoU parameters for both G-lite-Deeplabv3+ and the original Deeplabv3+ are presented in Table 3.

The MIoU parameter of an image serves as an effective measure of its segmentation accuracy, ranging from 0 to 1. A higher value signifies a better network segmentation effect on the target. By comparing the MIoU parameters of the proposed network with those of the conventional Deeplabv3+ network, it is observed that the improved network achieves a 6.4% higher segmentation effect on graphic elements, a 0.2% higher segmentation effect on the background, and an overall 3.3% higher segmentation effect. The analysis of MIoU parameters quantitatively demonstrates that the improved network exhibits superior segmentation capabilities.

## Performance comparison of different segmented networks

To further validate the efficacy of the proposed network in segmenting product packaging images, a comparative analysis is conducted with several existing models, namely UNet (*Kumar, Kumar & Lee, 2022*), ResUNet (*Zhou et al., 0000*), PSPNet (*Tan et al., 2021*), ACUNet (*Cao & Zhang, 2020*), and DeepLabv3+ with MobileNetv2 as the backbone (*Zhou, 2022*). The evaluation criteria for comparison encompass the intersection ratio (IoU), mean pixel accuracy (mPA), and frames per second (FPS). All experiments are performed on a self-developed dataset of product packaging images, employing identical software and hardware environments, as well as parameter settings. The comparison results are presented in Table 4.

**Table 4 Performance comparison of network models.**

| Network name | IoU | mPA | FPS |
|---|---|---|---|
| UNet | 0.916 | 0.880 | 35.3 |
| ResUNet | 0.937 | 0.909 | 12.4 |
| PSPNet | 0.917 | 0.896 | 52.6 |
| ACUNet | 0.921 | 0.901 | 42.9 |
| Deeplabv3+ | 0.936 | 0.911 | 32.26 |
| Ours | 0.965 | 0.967 | 39.4 |

As evident from the findings in Table 4, the proposed network demonstrates superior performance across all three evaluation metrics. By incorporating the attention mechanism and implementing further optimization, this network outperforms the DeepLabv3+ network, which solely utilizes MobileNetv2 for optimization. Specifically, compared to DeepLabv3+, this network achieves a 3.1% increase in IoU, a 6.2% increase in mPA, and a remarkable 22.1% increase in FPS, reaching a rate of 39.4/s. These advancements successfully meet the real-time demands for product packaging image segmentation.

## Comparison of different deployment methods

To validate the effectiveness of the CPS deployment mode in enhancing the segmentation efficiency of the G-lite-Deeplabv3+ network, real-time segmentation efficiency is compared with cloud computing deployment and edge computing deployment. The results are presented in Table 5.

The data processing time of the CPS deployment method, as implemented in this article, is found to be lower than that of cloud computing. For the same product packaging image segmentation task, the efficiency of the CPS deployment method is approximately 1.2 times higher than that of the edge computing method and about 2.8 times higher than that of the cloud computing method. This comparison highlights that the CPS deployment method employed in this article enables efficient product packaging image segmentation.

## Visualization of results

Figure 7 illustrates the network's convergence loss rate per epoch on the product packaging image dataset, comparing it with the network discussed in Section 4.1. Under the same number of training iterations, the G-Lite-DeepalV3+ network proposed in this article demonstrates superior convergence performance. It requires less time to reach the desired convergence level compared to other networks. Therefore, it can be inferred that the network in this article achieves the best segmentation model within the shortest training duration.

## CONCLUSION

In order to address the limitations of traditional segmentation methods in achieving real-time and accurate product packaging image segmentation in practical applications, this article introduces a lightweight method called G-Lite-DeepLabv3+. This approach replaces Xcept with MobileNetv2 as the feature extraction network and utilizes block convolution

**Table 5 Comparison of split time of different deployment methods.**

| Index | Cloud computing | Edge calculation | Cps |
|---|---|---|---|
| Split start time | 14:42:00 | 14:42:00 | 14:42:00 |
| Split end time | 14:44:24 | 14:45:58 | 14:45:58 |
| Transmission time/s | 0 | 0.5 | 0.55 |
| Calculation time/s | 0 | 184 | 118 |
| Statistical time/s | 191 | 0 | 0 |
| Working hours | 1005 | 422.5 | 356.55 |
| Efficiency comparison | Cloud computing: edge computing: cps =1:2.38:2.82 | | |

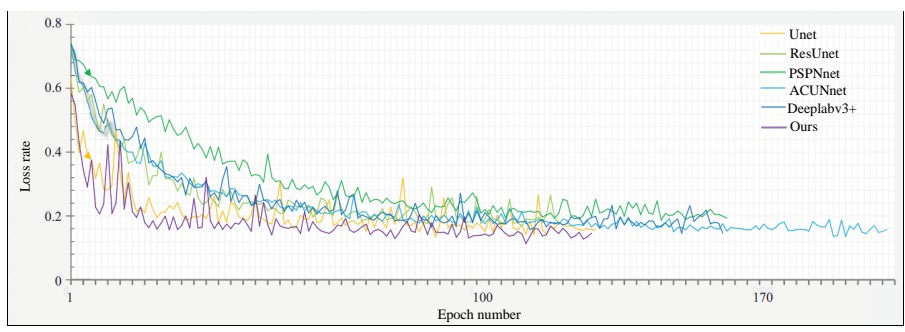

**Figure 7 Comparison of loss curve from training to convergence.**

instead of ordinary convolution within MobileNetv2 and the hollow space pyramid pool module. Additionally, two series-connected attention modules are incorporated to handle the high-level image features generated by the space pyramid pool module. G-Lite-DeepLabv3+ is integrated into the image processing module in the Cyber-Physical Systems (CPS) to enhance segmentation efficiency.

Experimental results demonstrate that the G-Lite-DeepLabv3+ network proposed in this article outperforms the control network in terms of both accuracy and efficiency, achieving MioU and IoU values of 0.939 and 0.965, respectively. This provides a solid foundation for the application of segmentation networks in product packaging image segmentation. However, the practical effectiveness of the presented method remains unknown. Therefore, future work will focus on further enhancing the network structure to achieve even more efficient and stable product packaging image segmentation in practical applications.

### Funding
The author received no funding for this work.

### Competing Interests
The author declares that there are no competing interests.

## Author Contributions

- Jiahao Wang conceived and designed the experiments, performed the experiments, analyzed the data, performed the computation work, prepared figures and/or tables, authored or reviewed drafts of the article, and approved the final draft.

## Data Availability

The dataset and code are in the Supplemental Files.

## Supplemental Information

Supplemental information for this article can be found online at http://dx.doi.org/10.7717/peerj-cs.1451#supplemental-information.

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
