# Peer review of "Packaging style design based on visual semantic segmentation technology and intelligent cyber physical system"

_PeerJ Computer Science, doi:10.7717/peerj-cs.1451_

## Round 0.1 · original submission · Major Revisions

Dear authors,
Your paper requires a couple of comments to be incorporated, therefore, please revise and re-submit the updated version.

Reviewer 1 ·

Basic reporting

This paper proposes a segmentation network based on G-lite-Deeplabv3+, and deploits it to Cyber Physical System to improve the accuracy and efficiency of product packaging image segmentation. Firstly, Mobilenetv2 is used to replace Deeplabv3+ feature extraction network, and packet convolution and attention mechanisms are introduced to process high-level semantic features, so as to improve the sensitivity of the network to useful features. Secondly, the image acquisition module, image processing module and output execution module are set respectively in CPS, and G-lite-Deeplabv3+ network is integrated into the image processing module in CPS to realize remote and real-time product packaging image segmentation in Cyber space. The following suggestions can help the author improve the manuscript.
 Before submitting a revision be sure that your material is properly prepared and formatted;
 If you are unsure, please consult the formatting instructions to authors that are given under the "Instructions and Forms" button in the upper right-hand corner of the screen;
 In the second paragraph of the introduction, the author lists a large number of existing studies, but the logical relationship between them is not clear. Some transitional statements can help readers better understand these works;
 Uncommon abbreviations (such as FPS, mPA) should be spelled out at first use. Do not include footnotes or references;
 Some references are too old and need to be replaced. The author had better use the latest literature in the last three years for citation;
 What is the optimization method of Lite-bottleneck?
 Add more description to Figure 2;
 The language expression of the conclusion part needs to be optimized, and the content of this part needs to supplement the limitations specifically;
 The author should give specific results relevant to the purpose; Avoid outcomes that are irrelevant to the purpose.

Experimental design

Please see report above

Validity of the findings

Please see report above

Reviewer 2 ·

Basic reporting

no comment

Experimental design

no comments

Validity of the findings

no comments

Additional comments

Aiming at the problem that traditional segmentation methods can not meet the real time and accuracy of product packaging image segmentation in practical application, a lightweight method G-lite-DeepLabv3+ is proposed. Experimental results show that the G-lite-DeepLabv3 + network proposed in this paper is superior to the control network in terms of accuracy and efficiency.
This study can provide some basis for improving the application of segmentation network in product packaging image segmentation task. This paper has some innovation, but it needs to be modified.

1. Try to set the problem discussed in this paper in more clear, write one section to define the problem;
2. I have not seen more details about how to determine the parameters of model network training (especially the selection of super parameters), but it is very important;
3. Add more analysis to Table 5, focusing on the comparison of data between different models and the underlying principles;
4. Legend description should be spontaneous, therefore, some description is not necessary, for example “The purple curve represents the method of this paper, under the same number of training rounds”;
5. The language of the conclusion needs to be further strengthened;
6. The description of the conclusion coincides with the abstract, and the author should focus on the outstanding contribution of the research and the application direction;
7. Most sentences contain grammatical and/or spelling mistakes or are not complete sentences;
8. The authors must have their work reviewed by a proper translation/reviewing service before submission; only then can a proper review be performed.

---

## Round 0.2 · accepted · Accept

Based on the recommendation of the experts, I am happy to let you know that your paper has been recommended for publication. Thank you for your fine contribution to our esteemed journal.

Reviewer 1 ·

Basic reporting

accept as it is.

Experimental design

No amendments required.

Validity of the findings

Agreed

Additional comments

No

Reviewer 2 ·

Basic reporting

No comments

Experimental design

None

Validity of the findings

No comments

Additional comments

I accept this work. Author Did a great job and carefully address all comments. I have no further comments.
Best of luck